# Environmental and Breed Risk Factors Associated with the Prevalence of Subclinical Mastitis in Dual-Purpose Livestock Systems in the Arauca Floodplain Savannah, Colombian Orinoquia

**DOI:** 10.3390/ani13243815

**Published:** 2023-12-11

**Authors:** Arcesio Salamanca-Carreño, Mauricio Vélez-Terranova, Diana Patricia Barajas-Pardo, Rita Tamasaukas, Raúl Jáuregui-Jiménez, Pere M. Parés-Casanova

**Affiliations:** 1Facultad de Medicina Veterinaria y Zootecnia, Universidad Cooperativa de Colombia, Villavicencio 500001, Colombia; 2Facultad de Ciencias Agropecuarias, Universidad Nacional de Colombia, Palmira 763531, Colombia; 3Unidad de Biotecnología, LABIPRESAN-UNERG, San Juan de los Morros 2301, Venezuela; 4Centro Universitario de Oriente, Universidad San Carlos de Guatemala, Chiquimula 20001, Guatemala; 5Institució Catalana d’Història Natural, 08001 Cataluña, Spain; pmpares@gencat.cat

**Keywords:** calving number, dairy farms, genetic traits, hand milking, prevalence level

## Abstract

**Simple Summary:**

Bovine mastitis is inflammation of the mammary gland. Subclinical mastitis (SCM) does not present visible changes in the udder or milk but does result in reduced production and alterations in milk. The aim of this study was to evaluate the environmental and breed risk factors associated with the presence of SCM in dual-purpose livestock systems in Arauca, Colombian Orinoquia. Milk samples were taken from the individual mammary quarters of 481 cows and the on-farm California Mastitis Test (CMT) was applied. Risk factors were determined by multiple logistic regression analysis. The response variable was the absence (0) or presence (1) of SCM. The environmental risk factors that were significantly associated (*p* < 0.05) with the presence of SCM were the number of cows and milk production. The Taurus-Indicus and composite breeds showed greater susceptibility to SCM compared to the Indicus predominance. The prevalence of SCM detected in this study is considered low compared to other studies in tropical regions.

**Abstract:**

The aim of this study was to assess the environmental and breed risk factors associated with the prevalence of subclinical mastitis (SCM) in cows in the dual-purpose livestock system of Arauca, Colombian Orinoquia. Milk samples were taken from 1924 mammary quarters, corresponding to 481 cows on 28 different farms, and the California Mastitis Test (CMT) was applied. Risk factors associated with SCM were determined using multiple logistic regression analysis. The response variable was the presence (1) or absence (0) of SCM. Breed was included as a genetic risk factor, and daily milk production, number of cows in production, lactation month, calving number, cow age, climatic period, and body condition were included as environmental risk factors. The analysis of the odds ratio (OR) of significant effects indicated that the factors significantly associated with the presence of SCM were the number of cows (OR = 2.29; *p* = 0.005), milk production (OR = 0.88; *p* = 0.045), and the Taurus-Indicus breeds (OR = 1.79; *p* = 0.009) and composite breed (OR = 3.95; *p* = 0.005). In this study, the occurrence of SCM was determined by the following risk factors: number of cows, milk production, and breed. Likewise, the highest prevalence seemed to occur on farms with less technological development and sanitary management of producers from the lowest socioeconomic stratum.

## 1. Introduction

Mastitis is a common disease in dairy farms that causes economic losses at local, regional, and global levels [1,2,3]. The disease is generally caused by microorganisms or trauma to the udder. It can be clinical when signs of disease and abnormal milk are visible or subclinical when there are no visible clinical signs [2,4] and the milk appears to be normal [5]. Subclinical mastitis (SCM) is considered the most predominant [6], being one of the most costly diseases in terms of milk production losses [7].

There are risk factors that predispose animals to the disease [8]. Among the risk factors associated with cows are genetic traits [8,9,10,11], the lactation stage, the calving number, the level of production [8,10,12], and the cow age [13]. At the milking level, the risk factors associated with the presence of SCM are inadequate washing of the udder and mammary quarters [13,14], cowshed hygiene [12], and handwashing [15]. Studies have indicated that the number of cows on the farm, the size of the farm, and the climatic period influence the presence of SCM [13,16,17]. Likewise, other studies have shown that the calving number, the farm system, the milking area, the region, and the herd are risk factors that significantly affect the prevalence of SCM [18].

The SCM influences the quality of milk, mainly in terms of composition, such as the reduction of phosphorus, calcium, protein, and fats, while increasing chlorine and sodium [19]. On the other hand, the consumption of raw milk with the presence of mastitis causes public health problems [20]. Understanding the risk factors is a prerequisite for improving udder health in a herd, region, or country [21].

At the farm level, a widely used and efficient method for the diagnosis of SCM is the California Mastitis Test (CMT), which has been used for decades under field conditions for the diagnosis of mastitis in cattle [22,23,24]. The CMT is a rapid, practical, and low-cost test with reliable results [25,26,27]. The precision of the CMT test has been evidenced in comparative tests of sensitivity and specificity [28]. Other studies have concluded that CMT is the most accurate test after somatic cell count and other laboratory tests; therefore, CMT is a reliable diagnostic method under farm conditions [23].

The CMT is a manual test that measures the quantity of somatic cells in milk generated by inflammatory processes [29,30]. Somatic cells migrate from the blood to milk as a response to infection [31]. The CMT identifies the inflammatory response based on the gel viscosity. The test contains a dye (bromocresol purple) that indicates pH changes that occur in milk as a result of inflammation [32].

The test does not provide a numerical result, but rather a categorical result, so any result above a vestigial reaction is considered suspicious, providing validity for the SCM diagnosis [33,34]. However, the test score with the number of somatic cells is valid for low-performing cows since they physiologically have a high level of somatic cells [35]. Other studies have determined, for individual cows, as CMT-positive, a score equal to or greater than 2, regardless of the absence of clinical symptoms, milk abnormalities, or the timing of milk production [18].

Milk production in Arauca (Colombia) comes mainly from the dual-purpose livestock system that is developed in tropical conditions, with low inputs and lower technological levels. The dual-purpose livestock system corresponds to groups of animals resulting from the crossing of Zebu cattle with European breeds. In the region, the health status of the udder of dairy animals is unknown, due to the few reports available in the area. Public order problems, the distance from urban centers, and the lack of passable access roads make access to the farms difficult. Therefore, microbiological examinations for the detection of SCM are difficult to perform. The practical applicability of CMT on the farm allows rapid diagnosis of SCM without the need for costly and time-consuming microbiological culture [36]. The study of risk factors for the prevalence of SCM allows us to measure the presence of an animal health problem in an area with a livestock attitude [19]. The aim of this cross-sectional study was to assess the environmental and breed risk factors associated with the prevalence of SCM in cows from the dual-purpose livestock system in the floodplain of Arauca, Colombian Orinoquia.

## 2. Materials and Methods

### 2.1. Study Area

The study was carried out in Arauca department, eastern Colombia (latitude: 7°5′5″ N; longitude: 70°45′32.7″ W) (Figure 1). The climatic regime of the region corresponds to a rainy period (May–October), with a relative humidity of 85% and an average ambient temperature of 30.1 °C, and a dry period (November–April), with a relative humidity of 65% and an average ambient temperature of 32.6 °C. The annual rainfall is less than 1500 mm [37,38]. 

The farms where the animals were sampled belong to small producers with a low technological level. The farms were chosen for convenience after a meeting with the Livestock Association Producers of the region. The average farm size was 65 hectares (ranging from 30 to 100 hectares). An average of 18 dairy cows per farm were found (range: 7–57 cows). All farms have productive and reproductive records. On the farms, milk production is measured every day (L/day/cow), and milking is performed by hand (once a day), with the calf’s presence, and is carried out under a covered cowshed or in open environments. The calf sucks before milking to stimulate milk let-down and then sucks again after milking. This is a normal practice in tropical dairy production. In most farms, animal feeding is based on extensive grazing systems with grasses, mainly *Brachiaria decumbes* and *Brachiaria arrecta*. On some farms, sodium chloride (white salt) is provided, and mineralized salts are provided to a lesser extent [36]. The destination of the milk is for self-consumption, homemade or artisan production, reception in collection centers, or direct sale to consumers. At the time of milking, udder washing of the cows was not observed. There was no health plan.

### 2.2. Evaluated Animals

Individual milk samples from 1924 mammary quarters belonging to 481 cows from the dual-purpose livestock system (Bos indicus x Bos taurus multiracial crosses), aged 3 to 10 years, with 1 to 8 calvings and 1 to 7 lactation months, were evaluated from 28 farms in 19 Territorial Division Centers.

Due to the lack of genealogical records for the animals (except the Gyrolando breed), the classification of cow breeds was based on their phenotype and the information provided by the producers, as recommended by other authors [39,40]: Taurus-Indicus (F_1_ cows Zebu x Holstein, Zebu x Brown Swiss, Zebu x Norman, and Zebu x Simental; *n* = 275)Indicus predominance (Cows with a phenotype higher than 50% Indicus of the Gyr, Guzerat, and Brahman breeds; *n* = 184)Composite breed (Gyrolando breed cows). The Gyrolando breed is a composite breed resulting from the cross between the Holstein breed (Bos taurus) and the Gyr breed (Bos indicus), with a racial pattern in grade 5/8 Holstein + 3/8 Gyr (*n* = 22).

Milk samples were taken from the cows in the morning (4:00 a.m. to 6:00 a.m.). All cows that were milked on the farm at the time of the visit were sampled. Only cows with functional mammary quarters and without antibiotic treatments during the last three months were included in the study. The samples were taken during the rainy period (May to October 2021) and the dry period (November to April 2022), following the recommendations available in the literature [18,26]. Mammary quarters were identified, and samples were taken in that order (Figure 2).

### 2.3. California Mastitis Test (CMT)

To sample each mammary quarter, 2 mL of milk was deposited in each receptacle of the plastic test paddle and mixed with 2 mL of CMT reagent (Mastit read) (Figure 2), following the standard procedure [29,30,32]. The mixture was homogenized for 10 s with circular movements [29] and the test results were then read with a 45° inclination. Interpretation of results was based on the standard procedure, following the CMT grades: 0 = negative, trace = possible infection, + = positive grade 1 (infected), ++ = positive grade 2 (infected), and +++ = positive grade 3 (infected) [29,30,32]. In the study, all results with some degree to CMT reaction were counted as SCM positive [27]. All tests were carried out by a single person.

### 2.4. Prevalence Determination

Prevalence is an indicator of existence or “stock”, since it considers all present cases, whether new or old, and refers to the number of cases in which a disease or infectious event occurs in a given place and time [41]. The cow-level prevalence (with at least one affected mammary quarter), prevalence at the level of total mammary quarters, and prevalence at the level of the mammary quarter by position were processed according to the following mathematical formulas [27,36,42]:Cow-level prevalence = (number of positive cows/total number of sampled cows) × 100Prevalence at the level of total mammary quarters = (total number of positive mammary quarters/total number of sampled mammary quarters) × 100Prevalence at the level of the mammary quarter by position = (total number of positive mammary quarters per position/total number of mammary quarters per position) × 100

Test data at the farm level were compiled and organized in Excel format for further analysis. The collected information included data related to risk factors: breed, cow age, calving number, lactation month, body condition, climatic period, number of cows in production, and daily milk production per cow. The body condition was subjectively evaluated at the time of taking the milk sample, on a scale from 1 to 5, where 1 is a very thin cow and 5 is a very obese cow [43].

### 2.5. Statistical Analysis

Frequency tables were estimated for the variables related to the number of positive cows (animals with any degree of positivity to SCM in any mammary quarter, as declared by the test), number of positive mammary quarters per cow (1 to 4), and positive mammary quarters by position (RP, LP, RA, and LA).

To determine the factors associated with SCM, a standard multiple logistic regression analysis was used, which is widely used for this study type [44]. The response variable was the SCM presence (1) or absence (0) obtained with the CMT.

To compare the variables of the number of cows and breed, dummy variables were created, and the categories with more than 26 cows (group 3) and animals with an Indicus predominance (group 2) were taken as a reference point, respectively. In the standard multiple logistic regression analysis, breed was included as a genetic risk factor, and the environmental risk factors included: cow age, daily milk production, lactation month, body condition, climatic period, number of cows in production, and calving number.

The following was the standard multiple logistic regression model used:N_i_ = log(π/(1 − π)) = M + β_1_X_1_ + β_2_X_2_ + β_3_X_3_ + β_4_X_4_ + β_5_X_5_ + β_6_X_6_ + β_7_X_7_ + β_8_X_8_ + εi
where:
N_i_ =i-th modeled probability of having SCM-positive animalsπ =odds ratio: (1 − π) probability of not having the presence of SCMM =slopeX_1_ =effect of the age variable (3 to 10 years)X_2_ =effect of the milk production variable (2 to 12 L)X_3_ =effect of the lactation month variable (1 to 7)X_4_ =effect of the body condition variable (1 to 5)X_5_ =effect of the climatic period variable (rainy and dry)X_6_ =effect of the number of cows variable (1 = 7 to 16 cows; 2 = 17 to 26 cows; 3 ≥ 26 cows)X_7_ =effect of the breed variable (1 = Taurus-Indicus; 2 = Indicus predominance; 3 = composite breed)X_8_ =effect of the calving number variable (1 to 8)εi =accumulated errorβ_i_ =regression coefficients associated with each independent variable.

The analyses were carried out using the statistical software InfoStat free version 2020 [45].

## 3. Results

A total of 1924 mammary quarters corresponding to 481 cows from the dual-purpose livestock system were sampled. The number of cows with mammary quarters affected by SCM, within the evaluated sample, is shown in Table 1. The cow-level prevalence indicates whether a cow presented the disease or if at least one of its mammary quarters were positive. Results showed that 151 cows were positive for SCM according to the CMT test, indicating a cow-level prevalence of 31.4% (151/481).

The SCM prevalence levels for total mammary quarters and mammary quarters by position are presented in Table 2. The CMT test showed a total mammary quarters prevalence level of 14.3% (2758/1924), while the highest mammary quarters prevalence level by position was for RP (16.0%) and RA (14.8%). 

Table 3 shows that the highest prevalence of SCM was observed with 7 to 16 cows in production (39.2%), in cows with 2 to 5 L/day (32.9%), in cows with more than 3 lactation months (33.1%), in cows more than 6 years of age (33.7%), in cows with a body condition of 2.5–3.5 (33.2%), and in cows with more than 4 calvings (38.9%). Regarding breed, the highest prevalence of SCM was observed in the composite breed (50.0%). On farms with 17–26 cows and >26 cows, the prevalence was similar (27.3% and 27.6%), and likewise between dry and rainy periods (31.1% and 31.4%).

Table 4 presents the factors that were significantly associated with the SCM presence in the evaluated systems. The standard multiple logistic regression analysis showed that most of the factors did not influence the positivity of SCM found with the CMT test, with the exception of the number of cows, milk production, and breed (*p* < 0.05).

The analysis of the odds ratio (OR) of the significant effects indicated that that cows raised in herds composed of 7 to 16 animals had a 2.29 times higher probability of presenting SCM than those raised in herds with more than 26 cows. In the case of milk production, for each unit increase in milk production, the odds of having sick animals with SCM vs. not sick with SCM was reduced by approximately 12%. With respect to breed, the Taurus-Indicus cows and composite breed cows were 1.79 and 3.95 more likely to present SCM than Indicus predominance cows.

It was observed that the number of cows per farm (7 to 16 cows) was a significant risk factor for the presence of SCM (39.2%; OR = 2.29; *p* = 0.005), compared to farms with a greater number of animals. Daily milk production was a factor that indicated a small statistical difference (*p* = 0.045; OR = 0.88; Wald LI_LS = 0.77–1.00) associated with the prevalence of SCM. Cows with 2 to 5 L of milk had the highest prevalence of SCM (32.9%), while the lowest prevalence was observed in cows with 6 to 12 L of milk (29.5%; Table 3 and Table 4).

Breed was a statistically significant factor associated with the prevalence of SCM. The highest prevalence (50%) was observed in composite breed cows (*p* = 0.005; OR = 3.95; Wald LI_LS = 1.50–10.40), while in Taurus-Indicus cows, a prevalence of 33.8% was observed (*p* = 0.009; OR = 1.79; Wald LI_LS = 1.15–2.79), compared to the Indicus predominance cows, which showed the lowest prevalence of SCM (25%).

## 4. Discussion

In this study, the CMT test detected a higher prevalence of SCM at the cow level than reported in other studies (20.2% and 20.5%) [14,46], but similar to other studies in tropical regions [4,47]. However, the results of the current study are considered low compared to other reports also in tropical regions [18,23,27,48]. On the study farms, milking was supported by the calf, and at the end of milking the residual milk was consumed, which may limit the growth of bacteria in the udder by reducing the presence of SCM [49]. The highest prevalence was observed in the RP and RA mammary quarters. The authors of this study observed that on the farms, during their rest period, the cows adopted a sternal decubitus position, often to the right side, which may allow the mammary gland to be more vulnerable to infection by environmental microorganisms. They also observed that manual milking usually began from the two right mammary quarters, or in a crossed way, starting with the RP and LA mammary quarters, which can facilitate the presence of mastitis in mammary quarters [36].

In this study, the number of cows per farm (7 to 16 cows) was a significant risk factor for the presence of SCM (OR = 2.29; *p* = 0.005), compared to farms with a greater number of animals (17 to 26 cows). These results may be associated with inadequate management practices in smaller farms, represented by poor milking practices, a milking place with dirt floors that facilitate environmental–animal contamination, insufficient infrastructure, poor handwashing of the milker and the udder, and not using sealant and gloves for milking. In addition, workers did not change or wash their clothes between milkings. These conditions were also evidenced in another study [50]. These results are similar to those reported in Southeast Asian countries [51] and in herds in Ethiopia [21]. However, the results of the present study differ from those reported in intensive milk production systems, where a higher cow density is associated with a greater risk of presenting SCM compared to semi-intensive systems (*p* = 0.017) [18]. Cows managed in intensive systems have a 10.3 times higher risk of suffering from SCM than those managed in systems with fewer cows [47]. In studies carried out in tropical climates of Bangladesh, the highest prevalence of SCM was for cows with semi-intensive management, stabled in areas with low grooming conditions, compared to those kept under intensive management (27.6% vs. 10.5%) [52].

Lower milk production indicated a small difference (OR = 0.88; *p* = 0.045) as a risk factor for the presence of SCM. However, the results were not conclusive, and future studies are required with a larger number of farms to obtain more reliable results. Direct observation by the authors infers that this risk factor seems to be associated with smaller farms, without sanitary management and a lower technological level. Furthermore, because these are lower production cows, farmers paid little attention to this form of mastitis. This result is similar to that reported in Sri Lanka, where the highest prevalence was found in daily milk production of 3 to 5 L/day [47]. In a study in Perú, it was found that hygiene before milking was a determining factor for the presentation of SCM in small producers [49]. However, in another study, milk/cow production was not considered as a risk factor for the prevalence of mastitis (*p* > 0.05) [18].

In this study, the Taurus-Indicus breeds (OR = 1.79; *p* = 0.009) and the composite breed (OR = 3.95; *p* = 0.005) were a significant risk factor associated with the presence of SCM, with a higher prevalence value compared to the Indicus predominance breeds. In a prevalence study in adapted Zebu breeds, Holstein Friesian crosses with local Zebu breeds, and Jersey breeds, a higher infection of the mammary gland was detected in Jersey cows (78.6%) and crossbreeds (51.9%) compared with the adapted Zebu breeds (16.7%) [53], which implies that the SCM presence is associated with high-yielding cows [54]. Various studies have reported that the breed effect is related to the presence of mastitis [55,56]. The higher SCM prevalence reflected in Taurus-Indicus cows and the composite breed suggests that milk production with manual milking and with the calf presence in the dual-purpose cattle system in Arauca with Indicus predominance cows may demonstrate a lower risk associated with SCM prevalence.

In another study, the authors found that breed and/or crossbreeding was not a risk factor in the prevalence of SCM, while the calving number (primiparous and multiparous), geographic region, and milk production (<10 L, 10–20 L, or >20 L) contributed significantly (*p* = 0.036) to the risk of SCM. In multiparous cows, the odds of having SCM were 2.51 times higher than the odds in first-calving cows [18]. It has also been reported that the prevalence of SCM was significant (*p* < 0.05) in European crosses (60.7%) compared to Sahiwal (55.5%) and local cattle (0%) [47].

Cows of breeds with high milk production are more susceptible to SCM [57] due to the size of the teats, which can easily become loose and allow the entry of pathogens, ultimately causing an infection [58]. In cows crossed with the Friesian breed, higher prevalence of SCM has been reported than in the local Zebu breed (56.4% vs. 26.8%) [28]. Likewise, higher levels of SCM have been reported for cross-breeds (*p* < 0.005) vs. local breeds, such as Zebu in Sri Lankan herds [52].

Finally, the highest prevalence was observed in smaller farms with a lower level of production, which differs from most studies. This indicates that the highest prevalence seemed to occur on farms of producers from the lowest socioeconomic stratum and with less technological development and sanitary management.

## 5. Conclusions

The occurrence of SCM in dual-purpose livestock systems of Arauca, Colombian Orinoquia, was determined by the risk factors of milk production, number of cows, and breed. In the current study, the prevalence of SCM detected at the cow level was considered low compared to other studies in tropical regions. Composite breeds and Taurus-Indicus under conditions of extensive management and manual milking were more susceptible to the presence of SCM. A higher prevalence of SCM was observed on farms with fewer cows and with milk production of 2 to 5 L. The results suggest that studies of risk factors associated with SCM should be carried out in dual-purpose livestock production systems in other regions.

## Figures and Tables

**Figure 1 animals-13-03815-f001:**
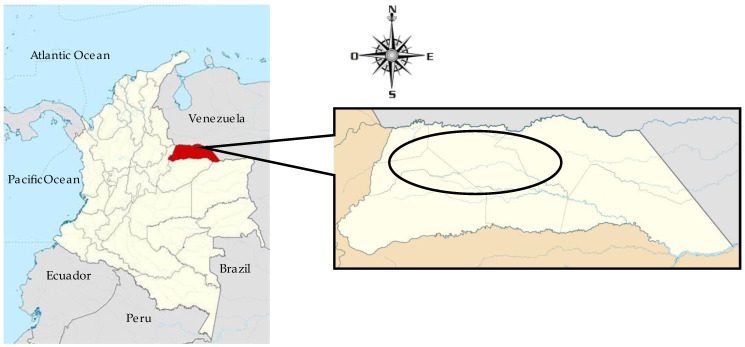
Red color: location of the department of Arauca, Colombian Orinoquia. Circle: area where the farms were sampled.

**Figure 2 animals-13-03815-f002:**
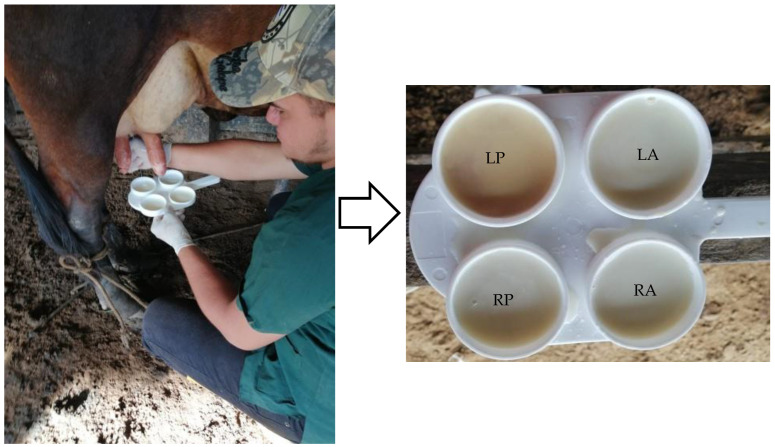
Mammary quarters’ positions for milk sampling. RP = right posterior, LP = left posterior, RA = right anterior, and LA = left anterior.

**Table 1 animals-13-03815-t001:** Number of cows with affected mammary quarters and cow-level prevalence in the sample evaluated in the dual-purpose livestock system of Arauca, Colombian Orinoquia.

Total sampled cows		481	%
Affected mammary quarters ^1^			
	1	85	17.7
	2	30	6.2
	3	14	2.9
	4	22	4.6
Positive cows		151	
Cow-level prevalence			31.4

^1^ = Cows with 1, 2, 3, and 4 affected mammary quarters.

**Table 2 animals-13-03815-t002:** Prevalence of SCM (%) at the levels of total mammary quarters and mammary quarters by position (RP, LP, RA, and LA) by the CMT test in a dual-purpose livestock system of Arauca, Colombian Orinoquia.

Total mammary quarters sampled		1924	%
Positives		275	
Prevalence at the level of total mammary quarters			14.3
Prevalence at the level of mammary quarters by position			
Mammary quarters	*n*	Positives	
RP	481	77	16.0
LP	481	64	13.3
RA	481	71	14.8
LA	481	63	13.1

*n* = Number of mammary quarters by position; RP = right posterior; LP = left posterior; RA = right anterior; LA = left anterior.

**Table 3 animals-13-03815-t003:** Prevalence of SCM (%) according to the number of cows in production, daily milk production, lactation month, cow age, climatic period, body condition, calving number, and breed, in Arauca, Colombian Orinoquia (descriptive analysis).

Variable	RP	LP	RA	LA	Positive Mammary Quarters	Positive Cows	Sampled Cows	%
Number of cows in production								
7–16 cows	26	20	23	27	96	60	153	39.2
17–26 cows	25	26	33	23	107	53	194	27.3
>26 cows	26	18	15	13	72	37	134	27.6
Daily milk production/cow								
2 to 5 L	41	33	41	33	148	78	237	32.9
6 to 12 L	36	31	30	30	127	72	244	29.5
Lactation month								
1–3 months	46	39	45	38	168	99	327	30.3
More than 3 months	31	25	26	25	107	51	154	33.1
Cow age								
3 and 4 years	17	17	22	16	72	41	153	26.8
5 years	29	21	21	24	95	50	153	32.7
6 or more years	31	26	28	23	108	59	175	33.7
Climatic period								
Rainy	73	57	66	56	252	128	411	31.1
Dry	4	7	5	7	23	22	70	31.4
Body condition								
2.5–3.5	40	32	32	27	131	69	208	33.2
3.6–4.5	37	32	39	36	144	81	273	29.7
Calving number								
1	13	12	16	13	54	33	116	28.4
2	27	19	21	22	88	49	165	29.7
3	15	16	15	12	58	26	92	28.3
>4	22	17	19	17	75	42	108	38.9
Breed								
Taurus-Indicus	43	43	44	36	166	93	275	33.8
Indicus predominance	27	16	24	24	91	46	184	25.0
Composite breed	7	5	3	3	18	11	22	50.0

RP = right posterior; LP = left posterior; RA = right anterior; LA = left anterior.

**Table 4 animals-13-03815-t004:** Odds ratios of the variables included in the standard multiple logistic regression for the associated factors with the SCM prevalence in dual-purpose livestock systems of Arauca, Colombian Orinoquia.

Regressor Variables	OR	Wald LI-LS (95%)	*p*-Value
Environmental factors			
Age	1.23	0.94–1.61	NS
Calvings	0.95	0.70–1.30	NS
Lactation	0.98	0.83–1.15	NS
Body condition	0.67	0.27–1.62	NS
Period	0.52	0.25–1.08	NS
Number of cows ^a^			
7 to 16	2.29	1.29–4.07	0.005
17 to 26	1.07	0.64–1.79	NS
Milk production	0.88	0.77–1.00	0.045
Genetic factors			
Breed ^b^			
Taurus-Indicus	1.79	1.15–2.79	0.009
Composite breed	3.95	1.50–10.40	0.005

OR: odds ratio; Wald LI-LS (95%): Wald confidence limits to 95%; ^a^ = reference point group 3 (>26 cows); ^b^ = reference point group 2 (Indicus predominance); NS = not significant.

## Data Availability

The data presented in this study are available on request from the second author. The data are not publicly available due to we are going to carry out other analyzes.

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
