# Peer review of "Environmental and Breed Risk Factors Associated with the Prevalence of Subclinical Mastitis in Dual-Purpose Livestock Systems in the Arauca Floodplain Savannah, Colombian Orinoquia"

_animals, 2023, doi:10.3390/ani13243815_

Round 1

Reviewer 1 Report (Previous Reviewer 2)

Comments and Suggestions for Authors

There are some details to improve in the description of the results and in the conclusions detailed in the document.

Author Response

First reviewer’s responses

Dear reviewer

The authors appreciate the insightful comments.

We attach the corrections and answers and were inserted into the text.

Comment

There are some details to improve in the description of the results and in the conclusions detailed in the document.

Response

Dear Reviewer: Corrections and responses were inserted into the text.

Reviewer 2 Report (New Reviewer)

Comments and Suggestions for Authors

The manuscript is basically carefully composed. Therefore, I support the publication of the manuscript. Before publication, it is worth correcting a few things in the manuscript. These are given below.

materials: please give us the exact number of animals according to their genoype. Each farm bred only one genotype or hold several genotypes, together? In latter case the effect of farm should be determined.

In the analysis model: Isupport to analyse the effect of farm on subclinicl mastitis

Author Response

Second reviewer’s responses

Dear reviewer

The authors appreciate the insightful comments.

We attach the corrections and answers and were inserted into the text.

 Comment

The manuscript is basically carefully composed. Therefore, I support the publication of the manuscript. Before publication, it is worth correcting a few things in the manuscript. These are given below.

materials: please give us the exact number of animals according to their genoype. Each farm bred only one genotype or hold several genotypes, together? In latter case the effect of farm should be determined.

Response

Table 3 shows the number of genotypes (sampled cows): Taurus-Indicus = 275; Indicus predominance = 184; Composite breed= 22

The three genotypes were not present in all farms. Some farms had one or two genotypes, so the farm effect was not considered but rather animals/farm.

Added in the text.

In the analysis model: Isupport to analyse the effect of farm on subclinicl mastitis

Response.

The effect of the farm was not analyzed because there were many and to include them in the analysis it was necessary to generate many Dummy variables. To avoid this, we group them by Territorial Division Centers. So, when they were grouped by Territorial Division Centers, the farms were also grouped by number of cows in the same way. Hence, the number of cows also corresponds to the farm effect. Furthermore, all farms have the same management characteristics.

Reviewer 3 Report (New Reviewer)

Comments and Suggestions for Authors

This paper does have merit in providing an initial idea on the incidence of subclinical mastitis in a farming situation which has difficulty of access. This will be important to make future improvements

Points to consider.

Lines 36-39. Use fewer decimal places for P values. Add an indication of the direction of the effect ie it was fewer cows and lower milk production. This is not necessarily what readers would expect, so important to mention.

L47 Word missing. Appears to be ?

L19. Unsure what is meant by “livestock attitude”.

L119. Production of what?

L141. In what order?

L145 to 148 move the description of the test to the introduction.

L156 Did they count “trace” as positive? Who did the tests – ie were different people involved? There is known variation in interpretation between operators and this needs to be mentioned.

L194. Farm of origin should have been included in the models as a random factor.

Table 2. Much is made of the analysis of the prevalence level by the quarter (Discussion L 268-275) but it is unlikely that the difference between 13.1 and 16.0% is significant. I suggest omitting this analysis.

L220 onwards, this sentence is too long and complicated.

Table 3. Mention in heading that this is a descriptive analysis.

L238 The parenthesis is in the wrong place. In any case, the percentage figures in this section are somewhat confusing. It would be better to just use the odds ratios, but these need to have their associated confidence limits added.

Minor points

L3.  In is repeated

L100 change de to the

L114, 115. Change suck to sucks

L151 should be agent

Spelling of litres

Comments on the Quality of English Language

Would benefit by being read by a native English speaker.

Author Response

Third reviewer’s responses

This paper does have merit in providing an initial idea on the incidence of subclinical mastitis in a farming situation which has difficulty of access. This will be important to make future improvements

Dear reviewer

The authors appreciate the insightful comments.

We attach the corrections and answers and were inserted into the text.

 Comment

Points to consider.

Lines 36-39. Use fewer decimal places for P values. Add an indication of the direction of the effect ie it was fewer cows and lower milk production. This is not necessarily what readers would expect, so important to mention.

Response. Added in the text.

L47 Word missing. Appears to be ?

Response. Corrected

L19. Unsure what is meant by “livestock attitude”.

Response. Region that, due to its agroecological conditions, livestock farming can be developed.

L119. Production of what?

Response. dairy production

L141. In what order?

Response. We don't understand your question. If it is the order of sampling of the mammary quarters, they are identified in figure 2 and table 2 as follows:

RP= Right Posterior; LP= Left Posterior; RA= Right Anterior; LA= Left Anterior

L145 to 148 move the description of the test to the introduction.

Response. The test description has been moved to the introduction.

L156 Did they count “trace” as positive? Who did the tests – ie were different people involved? There is known variation in interpretation between operators and this needs to be mentioned.

Response. Explained in the text. All tests were performed by a single person.

L194. Farm of origin should have been included in the models as a random factor.

Response.

The effect of the farm was not analyzed because there were many and to include them in the analysis it was necessary to generate many Dummy variables. To avoid this, we group them by Territorial Division Centers. So, when they were grouped by Territorial Division Centers, the farms were also grouped by number of cows in the same way. Hence, the number of cows also corresponds to the farm effect. Furthermore, all farms have the same management characteristics.

Table 2. Much is made of the analysis of the prevalence level by the quarter (Discussion L 268-275) but it is unlikely that the difference between 13.1 and 16.0% is significant. I suggest omitting this analysis.

Response. Ok, but according to our farm observations we want to inform the reader that these two mammary quarters present differences in the presence of SCM. It may be due to the way of milking, which is why we consider it of interest to the reader, especially the farmer and for him who milks.

L220 onwards, this sentence is too long and complicated.

Response. Corrected

Table 3. Mention in heading that this is a descriptive analysis.

Response. Corrected

L238 The parenthesis is in the wrong place. In any case, the percentage figures in this section are somewhat confusing. It would be better to just use the odds ratios, but these need to have their associated confidence limits added.

Response. Corrected in the text.

Minor points

L3.  In is repeated

Response. Corrected

L100 change de to the

Response. Corrected

L114, 115. Change suck to sucks

Response. Corrected

L151 should be agent

Response. Corrected

Spelling of litres

Response. Corrected

Reviewer 4 Report (New Reviewer)

Comments and Suggestions for Authors

The present article tries to uncover possible relations between sub clinical mastitis and environmental/genetic factors by means of a multiple linear regression model.

The subject area has been already investigated a lot and multiple articles are already present in literature but nonetheless the specific focus on the Colombian Orinoquia region, in my opinion, can justify this kind of specific analysis.

The study scheme can be considered correct, and the manuscript itself is quite clear and easy to read, however some major flaws need to be addressed especially in the discussion and reference sections.

Line 37: please check the p value of milk production.

Line 47: "and the milk appears to be" what? the sentence is incomplete.

Line 68-70: please rephrase the entire sentence, as of now is not clear at all.

Line 91: add "to" after "associated"

paragraph 2.1: consider providing in a supplementary table the locations of the farms considered in the study

Line 151: what do you mean by "gent"? maybe reagent?

Line 161: change round parenthesis with square parenthesis

paragraph 2.5: What specific multiple linear regression model has been used? standard? hierarchical? stepwise? please specify it in the section.

Line 198: It's either "carried out with the..." or "carried out using the..."

Line 239: remove ")." from the round parenthesis

Line 277: please check the p value, it's not the same as the table 4.

Line 282: please check the statement "workers should not change or wash their clothes between milkings" because to me workers SHOULD change their clothes between milkings in order to have better hygenic conditions.

Line 294: consider changing "least" with "lower".

Line 305: please add "of" after "prevalence".

Line 306: please consider adding "Rahularaj et al." before citation [48].

In the discussion section I have some doubts on the explanations that you give for the milk production risk factors. Your finding is completely opposite from what has been found in 95% of literature and the explanations that you give for this result are somewhat "weak". It is also necessary to note that the confidence interval for milk production OR comprehends the null value 1, and for this reason should not be regarded as protective factor or risk factor neither. Please revise the entire section in order to be more clear and understandable by the reader.

References: 18 of the references are written in spanish, please check if it's an error of format. If, instead, the articles are not written in english I strongly advise to search for english language articles for substitution. Also make sure that bacterial species are written in italics.

Comments on the Quality of English Language

Specific comments can be found in the "Comments and suggestions for the authors" section. Apart from that I would suggest to check the manuscript for minor grammar and syntax errors.

Author Response

Fourth reviewer’s responses

Dear reviewer

The authors appreciate the insightful comments.

We attach the corrections and answers and were inserted into the text.

 Comment

The present article tries to uncover possible relations between sub clinical mastitis and environmental/genetic factors by means of a multiple linear regression model.

The subject area has been already investigated a lot and multiple articles are already present in literature but nonetheless the specific focus on the Colombian Orinoquia region, in my opinion, can justify this kind of specific analysis.

The study scheme can be considered correct, and the manuscript itself is quite clear and easy to read, however some major flaws need to be addressed especially in the discussion and reference sections.

Line 37: please check the p value of milk production.

Response. Corrected

Line 47: "and the milk appears to be" what? the sentence is incomplete.

Response. Corrected

Line 68-70: please rephrase the entire sentence, as of now is not clear at all.

Response. Corrected

Line 91: add "to" after "associated"

Response. Corrected

paragraph 2.1: consider providing in a supplementary table the locations of the farms considered in the study

Response. We do not consider it appropriate to add the table in the text because the geographic information is not available because public order problems do not allow GPS equipment to enter the region.  The table of 28 farms and 19 Territorial Division Centers is shown here. We will add it as a complementary file.

Item

Farm

Territorial Division Centers

1

Betania

Nuevo Sol 

2

Delirios

Nubes 

3

Diamante

Caño Verde 

4

El Oriente

Caño Claro 

5

El Prado

Caño Claro  

6

El Progreso

Santo Domingo 

7

La Curva

 Santo Domingo 

8

La Esperanza

Las Gaviotas 

9

La Fortaleza

 Filipinas

10

La Gaviotas

Playa Rica 

11

La Guacharaca

Palestina 

12

La Pandorosa

Araguaney 

13

La Reserva

Caño Tigre 

14

Las Brisas (G)

La Guaira 

15

Las Brisas (NS)

Nuevo Sol 

16

Las Delicias

Jordán 

17

Los Cocos

Palestina 

18

Los Lagos

Palestina 

19

Los Placeres

Palestina 

20

Maravillal

Cesar 

21

Palmeras

Palestina 

22

Paratebueno

Galaxias 

23

Ponderosa

Independencia 

24

Porvenir

Flor Amarillo 

25

Prado

 Caño Verde

26

V/Alejandro

 Guaira

27

Valaguera

Independencia 

28

Villa Rosa

Arenosa 

Line 151: what do you mean by "gent"? maybe reagent?

Response. Corrected

Line 161: change round parenthesis with square parenthesis

Response. Corrected

paragraph 2.5: What specific multiple linear regression model has been used? standard? hierarchical? stepwise? please specify it in the section.

Response. Standard multiple linear regression model

Line 198: It's either "carried out with the..." or "carried out using the..."

Response. Corrected

Line 239: remove ")." from the round parenthesis

Response. Corrected

Line 277: please check the p value, it's not the same as the table 4.

Response. Corrected

Line 282: please check the statement "workers should not change or wash their clothes between milkings" because to me workers SHOULD change their clothes between milkings in order to have better hygenic conditions.

Response. Added: “In addition, workers do not change or wash their clothes between milkings”.

Line 294: consider changing "least" with "lower".

Response: corrected

Line 305: please add "of" after "prevalence".

Response. Corrected

Line 306: please consider adding "Rahularaj et al." before citation [48].

Response. Corrected

In the discussion section I have some doubts on the explanations that you give for the milk production risk factors. Your finding is completely opposite from what has been found in 95% of literature and the explanations that you give for this result are somewhat "weak". It is also necessary to note that the confidence interval for milk production OR comprehends the null value 1, and for this reason should not be regarded as protective factor or risk factor neither. Please revise the entire section in order to be more clear and understandable by the reader.

Response. Dear reviewer: The corrections have been made.

In the text it was added: “Lower milk production showed a small statistical difference (OR = 0.88; p = 0.045) as a risk factor for the presence of SCM. However, the results are not conclusive, future studies are re required with a larger number of farms to have more reliable results”.

And also, at the end of the text it says: “Finally, the highest prevalence is observed in smaller farms with a lower level of production, which differs from most studies. This indicates that the highest prevalence seems to occur on farms of producers from the lowest socioeconomic stratum and with less technological development and sanitary management”.

References: 18 of the references are written in spanish, please check if it's an error of format. If, instead, the articles are not written in english I strongly advise to search for english language articles for substitution. Also make sure that bacterial species are written in italics.

Response: Dear reviewer: most of the references are written in English. We consider that references in Spanish provide important information about work in SCM just as references in English do. The journal's standards do not require that all references be in English.

Comments on the Quality of English Language

Specific comments can be found in the "Comments and suggestions for the authors" section. Apart from that I would suggest to check the manuscript for minor grammar and syntax errors.

Response. English corrected

Round 2

Reviewer 3 Report (New Reviewer)

Comments and Suggestions for Authors

No further comments

Comments on the Quality of English Language

There are still a few minor errors in English so needs checking

Reviewer 4 Report (New Reviewer)

Comments and Suggestions for Authors

No further comments needed, I thank the authors for providing the needed corrections.

This manuscript is a resubmission of an earlier submission. The following is a list of the peer review reports and author responses from that submission.

Round 1

Reviewer 1 Report

Comments and Suggestions for Authors

See attachment

Comments on the Quality of English Language

Proofread the manuscript some sentences and paragraphs are not clear or too long, and that makes them confusing. 

Reviewer 2 Report

Comments and Suggestions for Authors

It is necessary to expand the information regarding how the information on some of the evaluated variables was obtained.

The results should be written with emphasis on those levels that represent a risk factor in order to capture greater attention and interest from the reader.

The discussion on some of the results obtained in the study should be expanded.

Reviewer 3 Report

Comments and Suggestions for Authors

The paper, titled “Environmental and breed risk factors associated with the prevalence of subclinical mastitis in in dual-purpose livestock systems in the Arauca floodplain savannah, Colombian Orinoquia.“ addresses an important and timely topic. I found the subject matter of the article fascinating and read the manuscript with great interest. The paper aligns well with the scope of the journal.

However, I believe that in the study shows some incomprehensible points:

·        The lack of proper conditions of udder and milking parlor cleaning, the hand milking system and inappropriate facilities may overestimate the results;

·        The subclinical mastitis predominance (SCM) was greater (p < 0.05) in this following occasions:

o   Farms with a fewer number of cows (7 to 16 cows);

o   Lower milk yield (2 to 5 L/d);

o   Composite breeds (Girolando);

However, the authors do not show any association among this factors, and this can lead to a misinterpretation about the real cause of a higher predominance of SCM.

So I have to suggest the rejection of the manuscript.